# Differential Amino Acid Uptake and Depletion in Mono-Cultures and Co-Cultures of *Streptococcus thermophilus* and *Lactobacillus delbrueckii* subsp. *bulgaricus* in a Novel Semi-Synthetic Medium

**DOI:** 10.3390/microorganisms10091771

**Published:** 2022-09-01

**Authors:** Andreas Ulmer, Florian Erdemann, Susanne Mueller, Maren Loesch, Sandy Wildt, Maiken Lund Jensen, Paula Gaspar, Ahmad A. Zeidan, Ralf Takors

**Affiliations:** 1Institute of Biochemical Engineering, University of Stuttgart, 70569 Stuttgart, Germany; 2Systems Biology, R&D Discovery, Chr. Hansen A/S, 2970 Hørsholm, Denmark

**Keywords:** microbial interactions, co-culture, *Lactobacillus bulgaricus*, *Streptococcus thermophilus*, milk, amino acid metabolism, metabolite exchange, flow cytometry, pH-dependent modeling, proteolytic activity

## Abstract

The mechanistic understanding of the physiology and interactions of microorganisms in starter cultures is critical for the targeted improvement of fermented milk products, such as yogurt, which is produced by *Streptococcus thermophilus* in co-culture with *Lactobacillus delbrueckii* subsp. *bulgaricus*. However, the use of complex growth media or milk is a major challenge for quantifying metabolite production, consumption, and exchange in co-cultures. This study developed a synthetic medium that enables the establishment of defined culturing conditions and the application of flow cytometry for measuring species-specific biomass values. Time courses of amino acid concentrations in mono-cultures and co-cultures of *L. bulgaricus* ATCC BAA-365 with the proteinase-deficient *S. thermophilus* LMG 18311 and with a proteinase-positive *S. thermophilus* strain were determined. The analysis revealed that amino acid release rates in co-culture were not equivalent to the sum of amino acid release rates in mono-cultures. Data-driven and pH-dependent amino acid release models were developed and applied for comparison. Histidine displayed higher concentrations in co-cultures, whereas isoleucine and arginine were depleted. Amino acid measurements in co-cultures also confirmed that some amino acids, such as lysine, are produced and then consumed, thus being suitable candidates to investigate the inter-species interactions in the co-culture and contribute to the required knowledge for targeted shaping of yogurt qualities.

## 1. Introduction

Dairy products have been a part of the human diet since ancient times [1]. Detailed identification and analysis of fermented milk products began in the twentieth century [2]. Efforts are ongoing to develop tools to examine lactic acid bacteria [3,4,5,6]. Yogurt, which is currently an important part of the cuisine of many cultures, will be a critical dietary component in the future. Therefore, the identification and determination of novel co-culture compositions that impart improved technological and organoleptic properties are active areas of research in the food industry [7]. *Streptococcus thermophilus* and *Lactobacillus delbrueckii* subsp. *bulgaricus* are the key species that drive yogurt production [2].

To meet the changing market demands, there is a need to understand the interaction between *S. thermophilus* and *L. bulgaricus* during milk fermentation and to make use of this knowledge to design improved food products [8]. Despite significant progress in the past, the current state of understanding still shows white spots [2].

In the last 15 years, metabolomics [9,10] and transcriptomics [11,12,13] have been widely applied to understand the physiology of *S. thermophilus* and *L. bulgaricus* in mono-culture and co-culture. Previous studies provide insights into the metabolites exchanged between the strains and elucidated the characteristic gene expression patterns. However, these datasets have provided a limited scope to assign contextual functionalities to metabolites [12,13,14].

Screening various combinations of *S. thermophilus* and *L. bulgaricus* strains in co-cultures is a time-consuming and costly process. Thus, only a small subset of all possible combinations and conditions has been investigated. To overcome this limitation, mathematical modeling approaches, such as community flux balance analysis, have been used to predict the performance of co-cultures [15]. Although the mathematical modeling approach enables the estimation of flux distributions in underdetermined systems, a minimum number of experimental measurements is required to limit the solution space. Additionally, the stoichiometry of interactions must be understood for the application of mathematical approaches. Both constraints require reliable and representative experimental datasets as a prerequisite for flux balance modeling [16].

Understanding of the complex metabolic interactions between *S. thermophilus* and *L. bulgaricus*, including the exchange of peptides and amino acids, is currently limited [2]. One key feature is the strong proteolytic activity of *L. bulgaricus*, which enhances the production of peptides and amino acids that become available for *S. thermophilus*, enabling growth [13]. However, some *S. thermophilus* strains exhibit proteolytic activity. Consequently, the question that arises is whether and what differences in this inter-species interaction exist when proteolytic and non-proteolytic *S. thermophilus* are combined with *L. bulgaricus* in co-cultures.

Acidification, a marker for lactic acid formation, may serve as an easy-to-follow readout once mono-cultures and co-cultures can be cultured under comparable conditions. Limited information is available on amino acid production and consumption [9] and potential amino acid depletion, which may trigger amino acid biosynthesis [12,13].

Milk is traditionally used as a growth medium for *S. thermophilus* and *L. bulgaricus* cultivations in the production of yogurt. *S. thermophilus* and *L. bulgaricus* produce lactic acid from lactose, which imparts an acidic taste and inhibits the growth of microbes, including *S. thermophilus* and *L. bulgaricus* [17,18]. However, milk composition is highly variable. Furthermore, milk comprises several complex ingredients that interfere with the sensitivity of analytical methods, such as high-performance liquid chromatography (HPLC) and mass spectrometry. Additionally, the acidification of milk leads to an increase in viscosity, which impairs the sensitivity of the analytical methods [19].

To overcome these intrinsic analytical barriers, this study developed a synthetic medium supplemented with amino acids (SMaa) to allow the growth of *S. thermophilus* and *L. bulgaricus* in mono-cultures, which enabled the analysis of individual growth characteristics. The synthetic medium may be supplemented with casein (SMcas) instead of amino acids to investigate the proteolytic abilities of *S. thermophilus* and *L. bulgaricus* in mono-cultures. The medium allows for investigation of the interactions between *S. thermophilus* and *L. bulgaricus* by excluding individual components that are likely to be exchanged. An important effect of the symbiotic relationship between *S. thermophilus* and *L. bulgaricus* is the faster acidification during milk fermentation [13]. Therefore, this study investigated this feature by co-cultivating the strains in SMcas.

This study presents a new medium and comparable datasets of *S. thermophilus* and *L. bulgaricus* in mono-culture and co-culture conditions, providing useful insights into essential amino acid production and consumption. Our results demonstrate that the patterns and levels of amino acid release and consumption in co-cultures are different from those of mono-cultures. These findings are essential for data-driven modeling and testing hypotheses on the induction of basic regulatory mechanisms in cells.

## 2. Materials and Methods

### 2.1. Strains and Cultivation Conditions

*Lactobacillus delbrueckii* subsp. *bulgaricus* strains (LB.1 = ATCC BAA-365, LB.2, LB.3, and LB.4) were provided by Chr. Hansen A/S and stored at −70 °C in Man–Rogosa–Sharpe (MRS) (69966 MRS Broth, Sigma-Aldrich Chemie GmbH, Steinheim, Germany) containing 20% (*v/v*) glycerol. For cultivation, the total cell suspension in the cryotube (1 mL) was transferred into 15 mL of MRS supplemented with 14.3 g L^–1^ lactose and incubated for 6–8 h at 40 °C [20,21,22,23]. After washing twice with 0.9% NaCl solution, the cell pellet was resuspended in 200 µL of 0.9% NaCl to inoculate the preculture containing SMaa. The preculture was cultured at 40 °C and gently stirred with a 10 mm magnetic bar at 400 rpm for 14–18 h until the pH was between 5 and 6. 

*Streptococcus thermophilus* strains (ST.1, ST.2, ST.3, and ST.4 = LMG 18311) were provided by the industrial partner (Chr. Hansen) and stored at −70 °C in M17 (56156 M-17 Broth, Sigma-Aldrich Chemie GmbH, Steinheim, DE, USA) containing 20% (*v/v*) glycerol. The cells in the cryotube were washed twice with 0.9% NaCl solution. Then, the cell pellet was resuspended in 200 µL of 0.9% NaCl to inoculate the preculture containing SMaa. The preculture was cultured at 40 °C and gently stirred with a 10 mm magnetic bar at 400 rpm for 2–6 h until the pH was between 5 and 6. 

Calculated amounts of biomass from *L. bulgaricus* and *S. thermophilus* precultures were washed twice with 0.9% NaCl solution and the cell pellets were resuspended in 200 µL 0.9% NaCl to inoculate the main culture. The main culture was carried out in SMaa or SMcas as indicated in Table 1. 

The preculture (SMaa) and main culture (SMaa or SMcas) were cultured in crimp-top serum bottles, which were pretreated by flushing with 80% N_2_ and 20% CO_2_ for 10 min at 400 rpm. Growth was monitored by measuring the optical density (OD) (λ = 600 nm) using a photometer (Amersham Bioscience, Ultrospec 10 cell density meter) or flow cytometry. 

### 2.2. Acidification Measurements

The pH was measured offline using a pH meter (SevenEasy^TM^, Mettler Toledo, Columbus, OH, USA) connected to a pH electrode (InLab Semi-Micro, Mettler Toledo, Columbus, OH, USA). 

### 2.3. Medium Preparation

#### 2.3.1. Complex Media

MRS (69966 MRS Broth, Sigma-Aldrich Chemie GmbH, Steinheim, Germany) was dissolved in Milli-Q water and the pH of the medium was adjusted to 6.5 using 2 M NaOH. Then, the medium was filtered using a 0.22-μm filter (ROTILABO^®^, PVD, Carl Roth GmbH & Co. KG, Karlsruhe, Germany) and sterile polysorbate 80 (CAS-Nr.: 9005-65-6, Sigma-Aldrich Chemie GmbH, Steinheim, Germany) was added according to the manufacturer’s instructions. 

M17 (56156 M17 Broth, Sigma-Aldrich Chemie GmbH, Steinheim, DE, USA) was prepared following the manufacturer’s instructions and autoclaved.

#### 2.3.2. Semi-Synthetic Medium

A sterile 5× basal solution containing di-potassium hydrogen phosphate, potassium dihydrogen phosphate, sodium acetate, ammonium citrate, manganese sulfate, iron(II) sulfate, and Tween 80 was prepared as indicated in Table 1. Sterile lactose, magnesium sulfate, urea, nucleobases, and amino acids were added to the solution. After the pH was set to 6.5 with 1 M HCl, trace elements, vitamins, calcium chloride, and casein were added. The serum bottle was sealed, crimped, and flushed with sterile 80% N_2_ and 20% CO_2_ for 10 min at 400 rpm. 

The casein stock solution was prepared in a beaker containing glass beads (3 mm in diameter), which were covered with a thin layer of 200 µL of Tween 80. Next, 100 mL of water containing 0.26 g L^−1^ CaCl_2_ was added, and the solution was stirred slowly overnight, followed by autoclaving for 5 min at 121 °C.

### 2.4. Cell Dry Weight (DW)

A glass vial (1 mL, VWR) was dried at 105 °C for at least 36 h, cooled at 20 °C for at least 1 h, and weighed. Aliquots of 1 mL of culture samples in SMaa were washed thrice with Milli-Q water (40 °C) in a 1.5-mL reaction tube (Eppendorf), resuspended in 300 µL of Milli-Q water, and transferred into a dried glass vial. The reaction tube was rinsed with 200 µL of Milli-Q water, and the water was transferred to the glass vial. The glass vial was dried at 105 °C for at least 36 h, cooled at 20 °C overnight in a desiccator, and weighed to calculate the cell dry weight. The correlation between optical density, flow cytometry data (events mL^−1^), and cell dry weight (g_DW_ L^−1^) was as follows: for LB.1, 1 g_DW_ L^−1^ = 0.17101671 × 10^−7^ * events mL^−1^ = 0.2527 × OD_600 nm_; for ST.1, 1 g_DW_ L^−1^ = 0.01970622 × 10^−7^ * events mL^−1^ = 0.2075 × OD_600 nm_; for ST.4. 1 g_DW_ L^−1^ = 0.043115 × 10^−7^ * events mL^−1^ = 0.243 × OD_600 nm_. 

### 2.5. Biomass Measurements Using Flow Cytometry

Samples for flow cytometry analysis were prepared as described previously [3]. The cell suspension (100 µL) was diluted 10-fold with Tris-HCl (1.3 M) EDTA (0.13 M) buffer (pH 8) and incubated for 10 min on a shaker (Eppendorf Thermomixer 5436, Hamburg, Germany) at 1200 rpm and 50 °C. Next, the cell suspension was incubated with 1× SYBR™Green I nucleic acid gel stain concentrate (Thermo Fisher Scientific, Waltham, MA, USA) for at least 10 min at 20 °C in the dark. The sample was filtered through a filter (Partec CellTrics^®^ 30 µM mesh filter size, Sysmex, Germany) into a polystyrene tube immediately before measurements and analyzed using a flow cytometer (BD Accuri™ C6; BD Bioscience, Franklin Lakes, NJ, USA) equipped with four fluorescence detectors (FL1 533/30 nm, FL2 585/40 nm, FL3 > 670 nm, and FL4 675/25 nm), two scatter detectors, a blue laser (488 nm), and a red laser (640 nm). Sterile Milli-Q water was used as the sheath fluid. The instrument performance was monitored weekly with BDTM CS&T RUO Beads. The threshold settings, FSC-H 500 and FL1-H 500, a limit of 25 μL, and the slow flow rate of 14 μL/min were used for the analysis of the samples.

The log-transformed FL1-A and FSC-H signals were used to enumerate the total number of events in a sample. The flow cytometry data of the first 10,000 events of the pure medium sample were used for a one-class support vector machine (SVM) classifier implemented in MATLAB^®^ using the command ‘fitcsvm’ to identify and remove signal from medium in samples. Additionally, the lower background data were removed using a linear line as the gate, resulting in a cleaned dataset. Linear correlations between cleaned flow cytometric data and the dry weight of cells cultured in SMaa were fitted to the measured data from LB.1, ST1, and ST.4 cultures (Appendix A). To determine the transferability of the linear correlation between flow cytometric data and cell dry weight from cells cultured in SMaa to cells cultured in SMcas, a 1:1 mixture (*v/v*) of both samples was prepared and measured using flow cytometry. Additionally, each sample was individually analyzed using flow cytometry. The calculated sum of the number of cell events cultured in SMaa and the number of cell events cultured in SMcas resulted in the same number of cell events in the measured mixture, indicating transferability (Appendix A).

Cell dry weight in co-cultures was calculated using the same method with determined transferability (Appendix A). The strain-specific cell events of *S. thermophilus* and *L. bulgaricus* in co-culture were estimated using manual classification or SVM classification depending on the pH of the sample (Appendix A). Manual classification was achieved by separating the flow cytometry data using a line (the log-transformed FSC-H signal was plotted against the log-transformed FL1-A signal and separated by a linear line). The data points above and below the line represent *L. bulgaricus* and *S. thermophilus*, respectively. Classification of strains in co-culture using SVM was achieved using the log-transformed FSC-H and FL1-A signals of mono-culture datasets. Background data were removed to optimize SVM parameters in MATLAB^®^ using the command ‘fitcsm’ (Appendix A). 

### 2.6. Quantification of Fermentation Products

The culture sample (0.5 mL) was centrifuged for 3 min at 20,000× *g* and 4 °C. The supernatant was stored at −70 °C. 

Sugars (lactose, glucose, galactose) and organic acids (lactate, succinate, formate) were quantified using the Agilent 1200 series HPLC system equipped with an RI detector [24]. Before analysis, the supernatant was incubated with 4 M NH_3_ and 1.2 M MgSO_4_ solutions, followed by an incubation for 15 min with 0.1 M H_2_SO_4_ to precipitate phosphate. Isocratic separation was achieved using a Rezex ROA organic acid H (8%) column (300 × 7.8 mm, 8 μm; Phenomenex) protected by a Phenomenex guard carbo-H column (4 × 3.0 mm) at 50 °C. The HPLC conditions were as follows: mobile phase, 5 mM H_2_SO_4_ solution; constant flow rate, 0.4 mL min^−1^. Absolute concentrations were obtained by standard-based external calibration, and rhamnose was used as an internal standard (1 g L^–1^) to correct measurement variability.

Amino acid concentrations were determined by an Agilent 1200 series instrument (Agilent Technologies) [24]. Bicratic separation was achieved by an Agilent Zorbax Eclipse Plus C_18_ column (250 by 4.6 mm, 5 µm), which was protected by an Agilent Zorbax Eclipse Plus C_18_ guard column (12.5 by 4.6 mm, 5 µm). After automatic precolumn derivatization with *ortho*-phthaldialdehyde, fluorometric detection (excitation at 230 nm and emission at 450 nm) was carried out. The elution buffer consisted of a polar phase (10 mM Na_2_HPO_4_, 10 mM Na_2_B_4_O_7_, 0.5 mM NaN_3_, pH 8.2) and a nonpolar phase (45% [*v/v*] acetonitrile, 45% [*v/v*] methanol). The quantification of amino acids was achieved by standard-based external calibration, and 4-aminobutanoic acid was used as an internal standard at 100 µM to correct for analyte variability.

### 2.7. Total Amino Acid Composition in the Supernatant

The culture sample (0.3 mL) was centrifuged for 3 min at 20,000× *g* and 4 °C. The supernatant was stored at −70 °C. The supernatant (200 µL) was incubated with 300 µL of 32% HCl at 100 °C for 24 h, cooled at 20 °C for at least 1 h, slowly mixed with 490 µL of 6.23 M NaOH, and stored at −20 °C until quantification of amino acid concentrations by HPLC analysis.

### 2.8. Calculation of Amino Acid Production Rates

Individual biomass-specific amino acid production rates *q_aa_* [mol g_DW_^−1^ h^−1^] were calculated for each amino acid in a differential manner at 1 h intervals. The average biomass *c_x_* [g_DW_ L^−1^] in the period Δ*t* [h], and the net amount of produced amino acids Δ*c_aa_* [mol L^−1^] (Equation (1)) were considered.
(1)$$q_{aa}=\frac{\Delta c_{aa}}{\frac{c_{x_{1}}+c_{x_{2}}}{2}\cdot \Delta t}$$

### 2.9. Fitting of Gaussian Models to pH-Dependent Amino Acid Production Rate

The release of amino acids strongly relies on enzymatic proteolysis. As the proteolytic activity depends on various enzymes with each contributing to an individual optimum pH [25,26], integral activities may be described by the superposition of Gaussian activity distributions. However, exact values for pH optima were not available. Additionally, *de novo* biosynthesis may occur, albeit to a minor extent. Consequently, the Gaussian model was considered a suitable proxy for the observed amino acid ‘production’ profiles. Parameter regression was achieved by fitting the pH-dependent *q_aa_* of the *L. bulgaricus* LB.1 mono-culture (Appendix A) using Equation (2) [27].
(2)$$q_{aa}=\overset{n}{\underset{i=1}{\sum }}a_{i}e^{\left[-{\left(\frac{pH-b_{i}}{c_{i}}\right)}^{2}\right]}$$
where *q_aa_* is the amino acid production rate [mol g_DW_^−1^ h^−1^]; *n* is the number of pH optima to fit; and *a*, *b*, and *c* are regression parameters coding for the shape of the curve. MATLAB ^®^ was used for fitting. The consideration of a single pH dependency is not always sufficient. Then, overlaying Gaussian models considering two pH optima were used to improve the model prediction quality (Appendix A).

### 2.10. Simulation of Amino Acid Concentrations

Changes of biomass, substrate, and product concentrations were described in a process model assuming batch operation modes by balancing biomass (Equation (3)), substrate (Equation (4)), and product (Equation (5)) within the system boundary.
(3)$$\frac{dc_{x}}{dt}=µ\cdot c_{x}$$
(4)$$\frac{dc_{s}}{dt}=-q_{s}\cdot c_{x}$$
(5)$$\frac{dc_{p}}{dt}=q_{p}\cdot c_{x}$$

The amino acid production kinetics were integrated into the process model to predict *c_aa_(t)*. The simulation time steps Δ*t* considered the mean pH and biomass values as indicated in Equation (6).
(6)$$c_{aa}=q_{aa}\cdot c_{x}\cdot \Delta t=\overset{n}{\underset{i=1}{\sum }}a_{i}e^{\left[-{\left(\frac{\frac{pH_{1}+pH_{2}}{2}-b_{i}}{c_{i}}\right)}^{2}\right]}\cdot \frac{c_{x_{1}}+c_{x_{2}}}{2}\cdot \Delta t$$

The feasibility of this approach was demonstrated for the mono-culture of *L. bulgaricus* LB.1 (Appendix A).

### 2.11. Uncertainty Analysis

Metabolite concentrations, pH, OD, flow cytometric data, and dry weight values were analyzed using Microsoft^®^ Excel. Mean and standard deviation were calculated using duplicates and triplicates (STABW.S) in Microsoft^®^ Excel. All experimental results are expressed as the mean of three biological replicates with experimental errors unless otherwise stated. 

## 3. Results

### 3.1. Medium Development

The main objectives for preparing the SMcas were as follows: (a) enabling the growth of both species in mono-culture, (b) enabling the growth of both species in co-culture, and (c) potential metabolites that may be exchanged [2,3,6,10,12,13,14,28,29] were excluded if growth was not affected. To obtain this medium, previously reported defined growth medium compositions of *S. thermophilus* [30,31] and *L. bulgaricus* [21,32] were compiled, resulting in a long list of constituents. This list was further reduced to achieve a lean growth medium to fulfil the demands (a–c). Medium acidification, which mirrors growth-coupled lactate formation, was used as a readout to verify the ability of the strains to grow with different modifications in the medium. Oleic acid, pyruvic acid, formic acid, orotic acid, niacin, spermine, ascorbic acid, thioglycolate, and 2′-deoxyguanosine, which were used in the growth medium by Chervaux et al. [32] but not by Grobben et al. [21], were excluded from the medium because they are not essential for the growth of *L. bulgaricus*. Additionally, we evaluated whether the addition of orotic acid is essential since it was considered to be an important component of the growth medium by Otto et al. [30] and Letort et al. [31]. Growth analysis of *L. bulgaricus* and *S. thermophilus* in the medium lacking orotic acid revealed culture acidification. The omission of biotin, thiamine, aminobenzoic acid, and thioctic acid did not result in the acidification in *S. thermophilus* culture but promoted the acidification in *L. bulgaricus* culture. Furthermore, urea was not excluded from the medium because it has previously been established that it increases the buffer capacity of the medium [31] and provides carbon dioxide and ammonia [3].

Studies using SMcas revealed the ability of three proteinase-positive *S. thermophilus* (ST.1, ST.2, and ST.3) strains and the four *L. bulgaricus* strains to acidify the medium. The proteinase-negative *S. thermophilus* ST.4 was not able to acidify SMcas and required access to free amino acids provided in SMaa (Appendix A).

Protocooperation between *L. bulgaricus* and *S. thermophilus* in co-culture has industrial relevance [2]. Co-culture benefits from the rapid exchange of metabolites, leading to accelerated acidification [13]. The effect of this protocooperation in the co-culture was observed in SMcas in the form of a faster acidification rate and a lower final pH (Appendix A).

### 3.2. Growth and Amino Acid Release in L. bulgaricus Mono-Culture

*L. bulgaricus* hydrolyzes amino acids from casein through its cell wall proteinase PrtB, which is complemented by other intracellular and extracellular peptidase activities [12,13,33,34]. Therefore, peptides and free amino acids can be utilized by *S. thermophilus*. Furthermore, amino acid depletion may upregulate amino acid biosynthesis in co-cultures [12,13]. Hence, a key step in understanding cellular responses to extracellular amino acid depletion is to monitor amino acid release and uptake.

*L. bulgaricus* LB.1 was cultured in SMcas as a mono-culture. The biomass of the culture increased from 0.05 to 0.6 g_DW_ L^–1^, whereas the pH decreased from 6.4 to 4.3 (Figure 1). Lactose was consumed, glucose was initially secreted (up to 1.4 mM) and then consumed, and galactose, lactate, formate, and succinate were produced (Appendix A) in the culture, indicating metabolic activity.

The following two patterns of amino acid release were observed (Figure 1): accumulation of alanine, serine, lysine, tyrosine, and valine from the beginning of culturing; other amino acids began to increase after 2 h. A previous study suggested that this lag time indicates cellular adaptation to casein through upregulation of proteolytic activity [9]. The initial release of tyrosine, arginine, serine, leucine, and valine indicates active proteolytic activity from the beginning of culturing as they might not be produced *de novo* from *L. bulgaricus* [13,35].

### 3.3. Growth and Amino Acid Release in Proteinase-Positive S. thermophilus Mono-Culture

The dynamics of amino acid release and uptake in the proteinase-positive *S. thermophilus* ST.1, amino acid concentrations were measured over a culturing period of 14 h (Figure 2). The following three distinct phases were identified: 0–5 h, increase of some amino acid concentrations but no change in biomass and pH; 5–10 h, acidification, biomass increase, and decrease of some amino acid concentrations while others kept increasing; 10–15 h, acidification, biomass decrease, and uptake and release of amino acids. The concentration of all analyzed amino acids increased at some time point. Additionally, the pH decreased from 6.6 to 4.7, whereas the biomass increased from 0.03 g_DW_ L^−1^ to 0.1 g_DW_ L^−1^ (Figure 2). Furthermore, 12 out of the 15 amino acids were consumed at some points in time. Moreover, the concentrations of some amino acids exhibited an oscillating release-consumption-release profile (e.g., serine and leucine). After 12 h, almost all lactose was consumed (30 mM), which was accompanied by the production of large amounts of glucose (22 mM) and lactate (30 mM) (Appendix A).

### 3.4. Growth and Amino Acid Release in the Co-Culture of Proteinase-Positive S. thermophilus and L. bulgaricus

Next, the amino acid concentrations in an *L. bulgaricus* LB.1—proteinase-positive *S. thermophilus* ST.1 co-culture were examined. The strains could grow in both SMcas (Figure 1 and Figure 2) and SMaa (Appendix A), indicating their ability to utilize casein and free amino acids. As shown in Figure 3, the concentration of all amino acids increased during cultivation at some point. The concentrations of aspartate, arginine, lysine, alanine, and isoleucine began to decrease after approximately 2 h. Meanwhile, the decrease in glycine concentration was delayed until 4 h. The following two phases were observed in amino acid release (Figure 3), growth, and acidification (Figure 4): 0–4 h, pH decreased from 6.4 to 4.7 while the growth of both strains was weak (Figure 4); 4–7 h, the biomass of *L. bulgaricus* increased from 0.05 g_DW_ L^−1^ to 0.22 g_DW_ L^−1^. Additionally, the consumption of 30 mM lactose, the production of 57 mM lactate, and the secretion (up to 10 mM) and uptake of glucose were observed (Appendix A).

### 3.5. Growth and Amino Acid Release in the Co-Culture of Proteinase-Negative S. thermophilus and L. bulgaricus

Next, the effects of replacement of proteinase-positive *S. thermophiles* ST.1 with proteinase-negative *S. thermophilus* ST.4 on the amino acid availability and the nutrient needs in the co-culture with *L. bulgaricus* LB.1 were examined. ST.4 could not grow in SMcas but could grow in SMaa (Appendix A). Therefore, a higher biomass fraction of *S. thermophilus* ST.4 was inoculated to avoid the anticipated overgrow of *L. bulgaricus*.

Figure 4B shows the following three phases: 0–2.5 h, increased biomass of *S. thermophilus* ST.4; 2.5–4 h, dominant growth of *L. bulgaricus* LB.1; 4–7 h, decreased biomass of *S. thermophilus* ST.4 even as *L. bulgaricus* LB.1 continued to grow. Hence, the presence of *L. bulgaricus* LB.1 enables the growth of *S. thermophilus* ST.4 in SMcas, which is consistent with previous findings [12]. Additionally, 25 mM of lactose was consumed and 58 mM of lactate was produced (Appendix A). Interestingly, lactose consumption severely slowed down after the growth stop of ST.4, while lactate formation continued. Furthermore, the concentrations of arginine (0–5 h), isoleucine (0–3 h), and lysine (0–7 h) decreased. Overall, the amino acid concentration in the proteinase-negative *S. thermophilus* ST.4—*L. bulgaricus* co-culture was lower than that in the proteinase-positive *S. thermophilus* ST.1—*L. bulgaricus* LB.1 co-culture.

### 3.6. Simulation of Amino Acid Concentrations to Compare Mono- and Co-Culture Cultivations

To indicate the changes in the amino acid profile when *S. thermophilus* was added to the *L. bulgaricus* culture, a Gaussian model of amino acid release dependent on pH and biomass was generated (see Methods). This model enables the simulation of the amount of amino acids released solely from *L. bulgaricus* in co-culture, which could not be identified in the mixed culture. Hence, the comparison between the simulation and measured data will indicate if the amino acid release activity differs between mono-culture and co-culture.

Amino acid profiles of *L. bulgaricus* mono-culture (Figure 1) were used to fit the Gaussian *q_aa_* models. Figure 3 compares the simulated amino acid profiles of *L. bulgaricus* with the measured amino acid profiles of the co-cultures, reflecting the results of the mixed culture interaction.

Generally, the amino acid concentrations in the proteinase-positive *S. thermophilus* ST.1—*L. bulgaricus* co-culture were higher than those in the simulated amino acid time courses of *L. bulgaricus* in mono-culture, with the exception of glycine and leucine.

By way of analogy, Figure 3 shows the difference between the measured amino acid concentrations in the *S. thermophilus* ST.4—*L. bulgaricus* co-culture and the simulated amino acid concentrations released from *L. bulgaricus*. Here, most of the measured amino acid profiles, except for alanine, tryptophan, and histidine, were lower than those of the simulated courses. This indicates increased uptake of amino acids, likely via the proteinase-negative *S. thermophilus* ST.4, which can only feed on amino acids and peptides released from *L. bulgaricus* but not from casein.

## 4. Discussion

### 4.1. Amino Acids Are Consumed by L. bulgaricus and S. thermophilus

In this study, amino acids were consumed by *L. bulgaricus* and *S. thermophilus* cultured in SMcas in both mono-culture (Figure 1 and Figure 2) and co-culture (Figure 3). This is in accordance with [22]. Amino acids were consumed even in the presence of peptide-bound amino acids (Appendix A). For example, lysine was consumed in the *S. thermophilus* ST.1—*L. bulgaricus* LB.1 co-culture after 4 h (Figure 3), although at least 230 µM of lysine bound to proteins and peptides was available (Appendix A).

This indicates that amino acid transporters are active and enable the strains to exchange amino acids that are produced through casein hydrolysis or biosynthesis [36,37]. Hence, it allows interaction [29,38,39]. Additionally, this enables the manipulation of *S. thermophilus* and *L. bulgaricus* cultivations in biotechnological processes by adding amino acids, such as lysine [40].

### 4.2. Amino Acids Can Accumulate in Cultivations with L. bulgaricus and S. thermophilus

*L. bulgaricus* LB.1 could accumulate all analyzed amino acids (Figure 1). Some of these amino acids accumulated from the beginning of culturing, indicating basal proteolytic activity although the strain was precultured under SMaa conditions. This suggests that *L. bulgaricus* LB.1 releases more amino acids from casein or/and produces amino acids than it is needed for growth and that amino acids become available for other strains [41]. The accumulation of amino acids indicates that extracellular peptidases are highly active [42], unusable amino acids are separated from peptides to gain posteriorly required amino acids, or proton-coupled amino acid secretion supports the maintenance of intracellular pH during acidification [43]. The poor release of amino acids in a *S. thermophilus* ST.1 cultivation reflects its low activity of peptidases [26,44].

### 4.3. Differences between Co-Cultures with Different S. thermophilus Strains

The proteinase-negative *S. thermophilus* ST.4—*L. bulgaricus* LB.1 co-culture yielded lower amino acid concentrations than the proteinase-positive *S. thermophilus* ST.1—*L. bulgaricus* LB.1 co-culture. This phenotype can be attributed to the increased growth of *S. thermophilus* ST.4 (Figure 4), which results in an enhanced demand for amino acids [45]. In addition, this observation is consistent with the lack of protease activity of *S. thermophilus* ST.4 (Figure 3). The depletion of arginine, lysine, and isoleucine observed in this study can upregulate peptidases or amino acid biosynthesis, which is consistent with the hypothesis of previous studies [9,12,13].

### 4.4. Co-Culture Is Not the Sum of Mono-Cultures

The proteinase-positive *S. thermophilus* ST.1—*L. bulgaricus* LB.1 co-culture yielded higher amino acid concentrations than the simulated concentration of amino acids released from only *L. bulgaricus* LB.1 (Figure 3). In particular, histidine was rarely released in the presumably histidine auxotroph *S. thermophilus* ST.1 mono-culture (Figure 2) [46] but was detected in high amounts in the *S. thermophilus* ST.1—*L. bulgaricus* LB.1 co-culture. The interaction between the two species may trigger metabolic changes in the strains, resulting in the rearrangement of metabolic fluxes [6,35,47]. Future studies must identify these co-culture triggers that serve as stimuli for basic metabolic adjustments. 

The amount of amino acid released from the co-culture was higher than the individual sums of the amounts of amino acid released from the mono-cultures. This might be a consequence of an upregulated proteolytic system in *L. bulgaricus* LB.1 and *S. thermophilus* ST.1. Alternatively, individual biosynthetic pathways might be stimulated in co-culture but not in mono-culture [46,48]. Previous studies have alluded to the upregulation of histidine biosynthesis [12,13]. 

### 4.5. Stimulatory Effects of Branched-Chain Amino Acid (BCAA) Depletion

Previous studies have hypothesized that BCAA availability is limited in the *S. thermophilus*—*L. bulgaricus* co-cultures due to the upregulation of BCAA permease in *L. bulgaricus* [13] and BCAA biosynthesis in *S. thermophilus* [12,13,49]. In this study, the levels of isoleucine, but not those of valine or leucine, were temporarily depleted in the co-cultures (Figure 3). Furthermore, the release of BCAA in the *L. bulgaricus* LB.1 mono-culture was similar to that reported in a previous study [9], which revealed that the proteolytic activity of *L. bulgaricus* promotes the excess release of BCAA from casein. In the LB.1 mono-culture, the final concentration of isoleucine (200 µM) was lower than that of valine (417 µM) and leucine (746 µM). This indicated isoleucine as a potential candidate for depletion. Additionally, low concentrations of isoleucine (up to 5 µM), leucine (up to 15 µM), and valine (up to 16 µM) were observed in the protease-positive *S. thermophilus* ST.1 mono-culture, indicating its ability to release BCAA from casein or biosynthesize BCAA [36,46]. However, the levels of isoleucine, leucine, and valine were lower than those in *L. bulgaricus*. Hence, isoleucine depletion is plausible and may result in the upregulation of BCAA permease in *L. bulgaricus* and BCAA biosynthesis in *S. thermophilus*, respectively.

### 4.6. Arginine and Lysine Depletion in Co-Cultures

Arginine and lysine concentrations were limited in the proteinase-negative *S. thermophilus* ST.4—*L. bulgaricus* LB.1 co-culture and oscillated in the proteinase-positive *S. thermophilus* ST.1—*L. bulgaricus* LB.1 co-culture (Figure 3). Previous studies [12,13] have reported the upregulation of arginine biosynthesis in *S. thermophilus* co-cultured with *L. bulgaricus*. Hence, our results support the hypothesis that low arginine concentrations might influence physiological responses [50], such as the upregulation of arginine biosynthesis in *S. thermophilus*.

## 5. Conclusions

In this work, we developed a synthetic medium that supports the growth of the dairy organisms *S. thermophilus* and *L. bulgaricus* in mono- and co-culture, which enables the quantitative monitoring of growth as well as substrate consumption and metabolite production dynamics. Amino acid release profiles in co-culture were not the sum of amino acid release profiles in mono-cultures. Additionally, the amino acid release profiles were not similar in co-cultures with different strain combinations. Amino acid depletion was observed in *S. thermophilus*—*L. bulgaricus* co-cultures, which may provide an explanation for the induced expression of proteolytic enzymes.

The uptake of several amino acids was observed during growth. Knowledge of co-culture-specific consumption rates for peptide and amino acid uptake along with release rates of amino acids provides a tool for determining yogurt quality and useful insights into cellular fitness for further strain and process optimization. Understanding cellular amino acid needs may enable a quantitative and detailed understanding of interactions in yogurt cultures. 

## Figures and Tables

**Figure 1 microorganisms-10-01771-f001:**
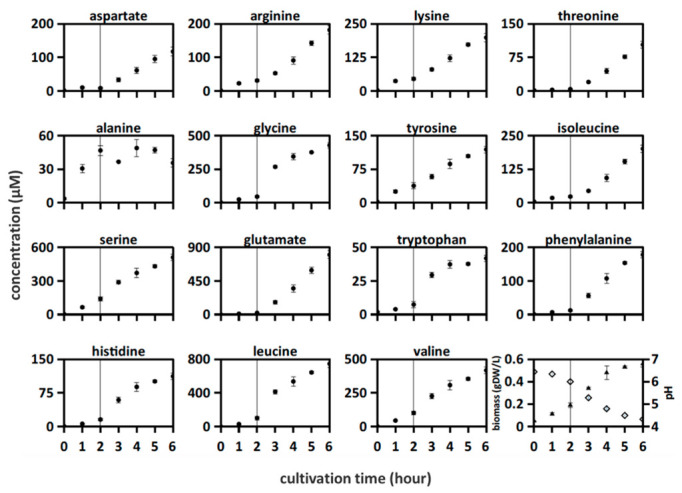
Amino acid concentrations were measured in *Lactobacillus bulgaricus* LB.1 culture in synthetic medium supplemented with casein (SMcas). The line indicates a change in increasing amino acid concentration profiles after 2 h. Downright: biomass (triangle) and pH (rhomb) measurements.

**Figure 2 microorganisms-10-01771-f002:**
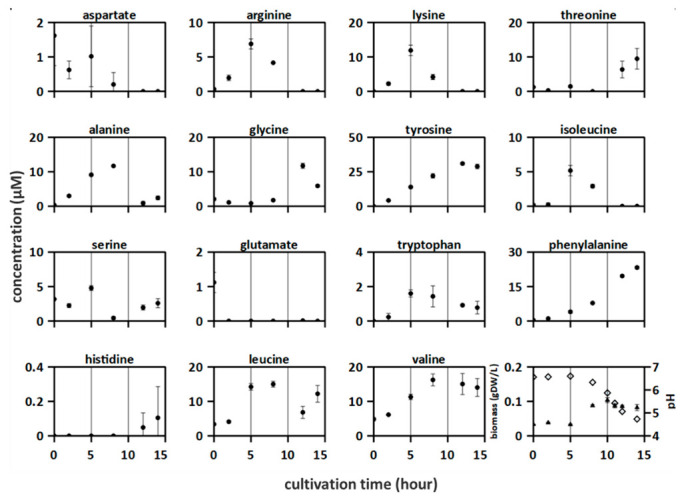
Amino acid concentrations were measured in proteinase-positive *S. thermophilus* ST.1 culture in synthetic medium supplemented with casein (SMcas). The lines indicate three phases according to the growth. Downright: biomass (triangle) and pH (rhomb) measurements.

**Figure 3 microorganisms-10-01771-f003:**
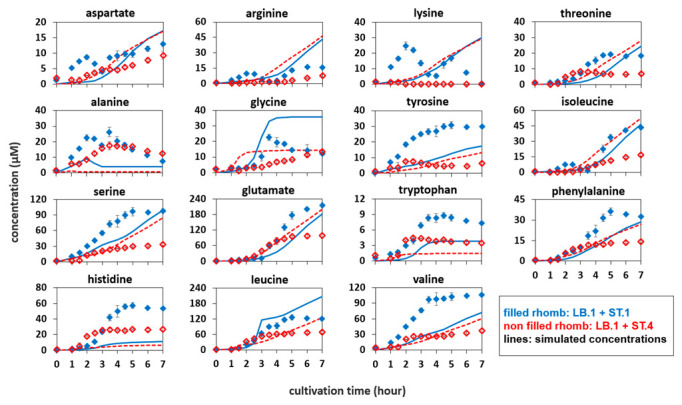
Amino acid concentrations in different co-cultures. (filled) *Lactobacillus bulgaricus* LB.1 co-cultured with proteinase-positive *Streptococcus thermophilus* ST.1 in synthetic medium supplemented with casein (SMcas). (non-filled) *L. bulgaricus* LB.1 co-cultured with proteinase-negative *S. thermophilus* ST.4 in SMcas. (line) Simulated amino acid concentration released from *L. bulgaricus* LB.1 in LB.1–ST.1 co-culture. (dashed line) Simulated amino acid concentration released from *L. bulgaricus* LB.1 in LB.1–ST.4 co-culture.

**Figure 4 microorganisms-10-01771-f004:**
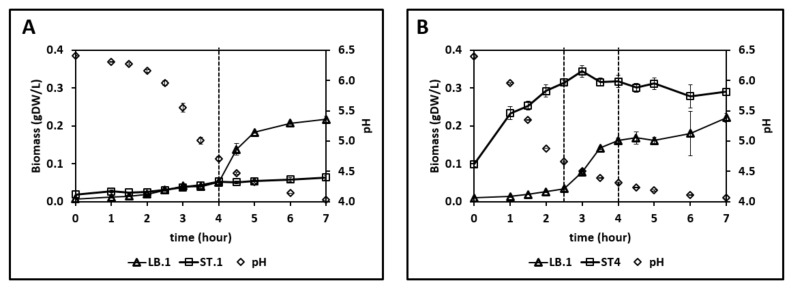
Strain-specific biomass profiles measured by flow cytometry and pH measurements in (**A**) LB.1–ST.1 (initial biomass fraction of 1:2 (LB:ST)) and (**B**) LB.1–ST.4 (initial biomass fraction 1:10 (LB:ST)) co-cultures in synthetic medium supplemented with casein (SMcas).

**Table 1 microorganisms-10-01771-t001:** Composition of synthetic medium (SM).

Category	Compound	Concentration [g L^−1^]	CAS Number
-	Di-potassium hydrogen phosphate	2.5	7758-11-4
	Potassium dihydrogen phosphate	3	7778-77-0
	Sodium acetate	1	127-09-3
	Ammonium citrate tribasic	0.6	3458-72-8
	Manganese sulfate monohydrate	0.02	10034-96-5
	Iron(II) sulfate heptahydrate	0.00132	7782-63-0
	Calcium chloride dihydrate	0.08745	10035-04-8
	Tween 80	1 mL L^−1^	9005-65-6
	D-Lactose monohydrate	15.75	10039-26-6
	Magnesium sulfate heptahydrate	0.2	10034-99-8
	Urea	0.12	57-13-6
nucleobases	Adenine	0.01	73-24-5
	Guanine	0.01	73-40-5
	Uracil	0.01	66-22-8
	Xanthine	0.01	69-89-6
vitamins	Biotin	0.0002	58-85-5
	Folic acid	0.0002	59-30-3
	Pyridoxal hydrochloride	0.001	65-22-5
	Riboflavin	0.0005	83-88-5
	Thiamine chloride hydrochloride	0.0005	67-03-8
	Nicotinamide	0.0005	98-92-0
	Cyanocobalamin	0.0005	68-19-9
	4-Aminobenzoic acid	0.0005	150-13-0
	D-Pantothenic acid hemicalcium salt	0.004	137-08-6
	DL-6,8-thioctic acid	0.0005	1077-28-7
trace elements	Ammonium molybdate tetrahydrate	0.0000037	12054-85-2
	Cobalt(II) chloride hexahydrate	0.000007	7791-13-1
	Boric acid	0.000025	10043-35-3
	Copper(II) sulfate pentahydrate	0.0000025	7758-99-8
	Zinc sulfate heptahydrate	0.0000029	7446-20-0
amino acids	L-Alanine	0.1	56-41-7
	L-Arginine	0.317	74-79-3
	L-Asparagine monohydrate	0.343	5794-13-8
	L-Aspartic acid	0.499	56-84-8
	L-Cysteine hydrochloride monohydrate	0.3	7048-04-6
	L-Glutamic acid	0.331	56-86-0
	L-Glutamine	0.29	56-85-9
	Glycine	0.16	56-40-6
	L-Histidine monohydrochloride monohydrate	0.273	5934-29-2
	L-Isoleucine	0.361	73-32-5
	L-Leucine	0.6	61-90-5
	L-Lysine	0.351	56-87-1
	L-Methionine	0.119	63-68-3
	L-Phenylalanine	0.34	63-91-2
	L-Proline	0.921	147-85-3
	L-Serine	0.359	56-45-1
	L-Threonine	0.3	72-19-5
	L-Tryptophan	0.102	73-22-3
	L-Tyrosine	0.12	60-18-4
	L-Valine	0.468	72-18-4
casein	Casein	2	9005-46-3

The SM contains all listed compounds, except amino acids and casein. SM supplemented with amino acids (SMaa) contains all listed compounds, except casein. SM supplemented with casein (SMcas) contains all listed compounds, except amino acids.

## Data Availability

Not applicable.

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
