# Peer review of "Differential Amino Acid Uptake and Depletion in Mono-Cultures and Co-Cultures of Streptococcus thermophilus and Lactobacillus delbrueckii subsp. bulgaricus in a Novel Semi-Synthetic Medium"

_microorganisms, 2022, doi:10.3390/microorganisms10091771_

Round 1
Reviewer 1 Report
The objective of the research is clear. The analyzes are well structured. This study would fill some white spots about co-cultures of Streptococcus thermophilus and Lactobacillus delbrueckii subsp. bulgaricus, which would be very meaningful. but there are some problems that need to be solved:
1. The title ′Differential amino acid uptake and depletion in mono-cultures and co-cultures of Streptococcus thermophilus and Lactobacillus bulgaricus′. ′Lactobacillus bulgaricus′ should be ′Lactobacillus delbrueckii subsp. bulgaricus′.
2. Line 39-40, ’Streptococcus thermophilus and Lactobacillus delbrueckii subsp. bulgaricus are the key species that drive yogurt production′. Line 47-50, ′Previous studies provide insights into the metabolites exchanged between the strains and elucidated the characteristic gene expression patterns. However, these datasets have provided limited scope to assign contextual functionalities to metabolites.′ Line 74-76, ′S. thermophilus and L. bulgaricus produce lactic acid from lactose, which imparts an acidic taste and inhibits the growth of microbes, including S. thermophilus and L. bulgaricus.′ Add the references.
3. The ratio of S. thermophilus and L. bulgaricus strains should be offered, such as 1:1 or 1:2, and the total number of initial colonies in the different experimental groups should be consistent.
4. In the part of 2. Materials and Methods, references or standards should be provided for all analytical methods.
5. In the part of 2.1. Strains and cultivation conditions (Line 98-112), the lactose was added into MRS, why? Why is the incubation temperature 40 ℃? generally, the incubation temperatures of S. thermophilus and L. bulgaricus are 37 or 42 ℃. What is meaning of 400 rpm? Shake culture? The cultures of S. thermophilus and L. bulgaricus do not need shake culture. The preculture was cultured until the pH was between 5 and 6, why?
6. In Table 1, the concentrations of casein and lactose are 2 and 15.75 g/L, why?
7. In the part of 2.6. Quantification of fermentation products (Line 205-213), only rhamnose was used as an internal standard to correct measurement variability, which was not suitable for quantification of organic acids. Using the external method by structuring the standard curves to quantify the concentrations of sugars and organic acids is more accurate. Besides, why only measure the concentrations of lactose, glucose, galactose, lactate, succinate and formate?
8. Line 268, could you explain the reason why the preparation of the SMcas could exclude metabolites that may be exchanged?
9. In the part of 2.11. Uncertainty analysis, it is more rigorous to conduct ANOVA to compare the data of different experimental groups.
10. Why LB.1 and ST.1 were chosen for co-cuture? why not LB.2, LB.3, or ST.2, ST3?
11. In the part of 4. Discussion, too few references are cited, and the discussion about the results of experiments is not in-depth enough and needs to be supplemented.
12. In part of 5. Conclusion, references should not appear in this part. The author should condense the main conclusions of this study. Besides, too many passages appeared in Introduction part.
Author Response
Cover letter for revision of the manuscript with the title: “Differential amino acid uptake and depletion in mono-cultures and co-cultures of Streptococcus thermophilus and Lactobacillus bulgaricus”
Reviewer 1
The objective of the research is clear. The analyzes are well structured. This study would fill some white spots about co-cultures of Streptococcus thermophilus and Lactobacillus delbrueckii subsp. bulgaricus, which would be very meaningful. but there are some problems that need to be solved
- We thank you very much for your well-grounded questions and your efforts to clarify our study.
- The title ′Differential amino acid uptake and depletion in mono-cultures and co-cultures of Streptococcus thermophilus and Lactobacillus bulgaricus′. ′Lactobacillus bulgaricus′ should be ′Lactobacillus delbrueckii subsp. bulgaricus′.
- Thank you very much for your indication. We changed the title.
- Line 39-40, ’Streptococcus thermophilusand Lactobacillus delbrueckii subsp. bulgaricus are the key species that drive yogurt production′. Line 47-50, ′Previous studies provide insights into the metabolites exchanged between the strains and elucidated the characteristic gene expression patterns. However, these datasets have provided limited scope to assign contextual functionalities to metabolites.′ Line 74-76, ′S. thermophilus and L. bulgaricus produce lactic acid from lactose, which imparts an acidic taste and inhibits the growth of microbes, including S. thermophilus and L. bulgaricus.′ Add the references.
- We included references. Please see manuscript.
- The ratio of S. thermophilus and L. bulgaricus strainsshould be offered, such as 1:1 or 1:2, and the total number of initial colonies in the different experimental groups should be consistent.
- We calculated initial biomass fraction (gram biomass LB per liter / gram biomass ST per liter). Please see manuscript, figure 4.
- In the part of 2. Materials and Methods, references or standards should be provided for all analytical methods.
- We included references. Please see manuscript.
- In the part of 2.1. Strains and cultivation conditions (Line 98-112), the lactose was added into MRS, why? Why is the incubation temperature 40 ℃? generally, the incubation temperatures of S. thermophilusand L. bulgaricus are 37 or 42 ℃. What is meaning of 400 rpm? Shake culture? The cultures of S. thermophilus and L. bulgaricus do not need shake culture. The preculture was cultured until the pH was between 5 and 6, why?
Thank you very much for your interest in the methods.
- The incubation temperature (40°C) was chosen based on literature and was assumed to be a combination of incubation temperature of ST and LB.
- ‘400 rpm’ indicates stirring (rounds per minute) by a magnetic stirrer preventing sedimentation of cells and casein due to low viscosity of the medium
- Fig S4 indicates that exponential growth is present between pH 6 to 5 indicating high metabolic activity which is a prerequisite for the subsequent quantification amino acids production and consumption.
- In Table 1, the concentrations of casein and lactose are 2 and 15.75 g/L, why?
- initial Casein and lactose concentrations in the medium were chosen to prevent protein or sugar overload of samples which might interfere with analytical methods. However, table s1 shows that sufficient peptides are available (see also discussion 4.1). Fig. S3, S5, and S7 indicate no limitation for lactose.
- The MRS was purchased from Sigma and did not contain lactose.
- In the part of 2.6. Quantification of fermentation products (Line 205-213), only rhamnose was used as an internal standard to correct measurement variability, which was not suitable for quantification of organic acids. Using the external method by structuring the standard curves to quantify the concentrations of sugars and organic acids is more accurate. Besides, why only measure the concentrations of lactose, glucose, galactose, lactate, succinate and formate?
- Rhamnose was added to each sample to normalize measurement variability due to changes between samples.
- Quantification of metabolite concentration was achieved by prior (and subsequent) establishment of a correlation between peak area and concentration.
- Other sugars and organic acids were measured in some experiments. But detected amounts were very low compared to presented metabolite concentrations. Therefore, this study focused on readily apparent uptake and consumption.
- Line 268, could you explain the reason why the preparation of the SMcas could exclude metabolites that may be exchanged?
- Exclusion of metabolites was based on literature. Please see manuscript, result section, 3.1.
- In the part of 2.11. Uncertainty analysis, it is more rigorous to conduct ANOVA to compare the data of different experimental groups.
- Thank you very much for this useful information. We will consider ANOVA in future studies.
- Why LB.1 and ST.1 were chosen for co-cuture? why not LB.2, LB.3, or ST.2, ST3?
- 1 and ST.1 show highest acidification in medium with casein indicating its high proteolytic activity (fig. S1). Therefore, the LB.1 - ST.1 co-culture is opposed to LB.1 – (protease-negative) ST.4 co-culture .
- In the part of 4. Discussion, too few references are cited, and the discussion about the results of experiments is not in-depth enough and needs to be supplemented.
- We extended the discussion and added related the discussed aspects to literature. Please see manuscript.
- In part of 5. Conclusion, references should not appear in this part. The author should condense the main conclusions of this study. Besides, too many passages appeared in Introduction part.
- We revised the conclusion. Please see manuscript.

Reviewer 2 Report
Excellent work !!!
I think that the words "new medium" should be included in the title.
Introduction: A few lines perhaps on similar reesearch performed by others??
Results: the text in some points feel more like discussion (lines 271-285 for example).
Author Response
Reviewer 2
Excellent work !!!
- Thank you very much for the compliment.
I think that the words "new medium" should be included in the title.
- Thank you very much for your indication. We changed the title.
Introduction: A few lines perhaps on similar reesearch performed by others??
- Please see line 51.
Results: the text in some points feel more like discussion (lines 271-285 for example).
- We changed the wording.

Reviewer 3 Report
This manuscript describes a thorough study of the interaction between two bacterial species commonly used in yoghurt manufacture. This follows the trend towards the analysis of multi-species in a variety of environments which trend towards what is happening in the real world more than focusing on individual species. The results will be of interest to those in the dairy industry.
Author Response
Thank you very much for revising our manuscript.
Round 2
Reviewer 1 Report
The authors have answered all the comments carefully. I recommend it to be accpted in this version.